Exogenous melatonin improves growth in hulless barley seedlings under cold stress by influencing the expression rhythms of circadian clock genes

Chang Tianliang
Zhao Yi
He Hongyan
Xi Qianqian
Fu Jiayi
Zhao Yuwei zhaoyw@nwu.edu.cn
1 Provincial Key Laboratory of Biotechnology of Shaanxi Province , Xi’an , China
2 Life Sciences School of Northwest University , Xi’an , China
3 Key Laboratory of Resource Biology and Biotechnology in western China (Ministry of Education) , Xi’an , China
Silva Pedro
Electronic publication date: 2021 Jan 22
Publication date: 2021
Volume: 9
Electronic Location ID: e10740
Received 2020 Aug 18; Accepted 2020 Dec 18
Copyright: ©2021 Chang et al.
Copyright year: 2021
Copyright holder: Chang et al.
License: This is an open access article distributed under the terms of the Creative Commons Attribution License, which permits unrestricted use, distribution, reproduction and adaptation in any medium and for any purpose provided that it is properly attributed. For attribution, the original author(s), title, publication source (PeerJ) and either DOI or URL of the article must be cited.
License URL: https://creativecommons.org/licenses/by/4.0/

Keywords: Melatonin, Circadian, Rhythm, Hulless barley, Cold stress

Funding: National Natural Science Foundation of China 31470329 Research Project of Provincial Key Laboratory of Shaanxi 17JS127 Research Project of Key Laboratory of Resource Biology and Biotechnology in Western China (Ministry of Education) ZSK2018005 This work was supported by grants from the National Natural Science Foundation of China (Grant number. 31470329), the Research Project of Provincial Key Laboratory of Shaanxi (Grant number: 17JS127) and the Research Project of Key Laboratory of Resource Biology and Biotechnology in Western China (Ministry of Education) (Grant number: ZSK2018005). The funders had no role in study design, data collection and analysis, decision to publish, or preparation of the manuscript.

==============================
Background

Melatonin is a hormone substance that exists in various living organisms. Since it was discovered in the pineal gland of cattle in 1956, the function of melatonin in animals has been roughly clarified. Nevertheless, in plants, the research on melatonin is still insufficient. Hulless barley (Hordeum vulgare L. var. nudum hook. f.) is a crop that originates from cultivated barley in the east, usually grown on the Qinghai-Tibet Plateau, becoming the most important food crop in this area. Although the genome and transcriptome research of highland barley has gradually increased recently years, there are still many problems about how hulless barley adapts to the cold climate of the Qinghai-Tibet Plateau.

Methods

In this study, we set three temperature conditions 25°C, 15°C, 5°C hulless barley seedlings, and at the same time soaked the hulless barley seeds with a 1 µM melatonin solution for 12 hours before the hulless barley seeds germinated. Afterwards, the growth and physiological indicators of hulless barley seedlings under different treatment conditions were determined. Meanwhile, the qRT-PCR method was used to determine the transcription level of the hulless barley circadian clock genes under different treatment conditions under continuous light conditions.

Results

The results showed the possible mechanism by which melatonin pretreatment can promote the growth of hulless barley under cold stress conditions by studying the effect of melatonin on the rhythm of the circadian clock system and some physiological indicators. The results revealed that the application of 1 µM melatonin could alleviate the growth inhibition of hulless barley seedlings caused by cold stress. In addition, exogenous melatonin could also restore the circadian rhythmic oscillation of circadian clock genes, such as HvCCA1 and HvTOC1, whose circadian rhythmic phenotypes were lost due to environmental cold stress. Additionally, the results confirmed that exogenous melatonin even reduced the accumulation of key physiological indicators under cold stress, including malondialdehyde and soluble sugars.

Discussion

Overall, these findings revealed an important mechanism that exogenous melatonin alleviated the inhibition of plant vegetative growths either by restoring the disrupted circadian rhythmic expression oscillations of clock genes, or by regulating the accumulation profiles of pivotal physiological indicators under cold stress.

Introduction

Melatonin is a hormone that is ubiquitously distributed in living organisms including animals, plants and algae (Tekbas et al., 2008). In animals, melatonin is involved in the regulatory process of many physiological behaviors through the synchronization of the circadian rhythm, such as the regulation of sleep-wake cycle, blood pressure, food intake, mood, locomotor activities, the number of immune cells in blood and reactive oxygen species scavenging processes (Nawaz et al., 2015). In plants, it is reported to have a wide range of functions involved in a number of physiological processes during either the vegetative or the reproductive growth stages of plants, such as promotion of seed germination, acceleration of seedling growth and regulation of plant senescence and cell death (Agathokleous, Kitao & Calabrese, 2019). In addition, the application of low doses of exogenous melatonin (<10 µM) to the culture medium can promote the growth of maize seedlings by increasing the photosynthetic efficiency (Zhao et al., 2015). Similarly, the inhibition of the expression of the SNAT gene which encodes an N-acetyltransferase enzyme involved in melatonin synthesis, also reduced the production of melatonin, and either retarded the growth of seedlings or their final yield in rice. Meanwhile, SNAT-deficient rice seedlings with reduced endogenous melatonin levels also exhibit a phenotype that is more sensitive to abiotic stresses, such as cold and salt stress than wild type (Byeon & Back, 2016). Under various abiotic stress conditions, such as cold, drought, salt stress, etc., melatonin plays an important role in inhibiting plant ROS production after stress (Turk et al., 2014; Zhang et al., 2014). In Malus hupehensis, melatonin reduces ROS-induced oxidative damage by directly scavenging H2O2 and enhancing the activity of antioxidant enzymes (such as ascorbate peroxidase, catalase and peroxidase) (Li et al., 2012). Additionally, it has been reported that exogenous melatonin can enhance the freezing resistance of Arabidopsis thaliana by inducing the expression of CBF/DREB (Shi, Wei & He, 2016).

Due to the day-night cycle caused by the Earth’s rotation, plants naturally undergo a circadian rhythm with a period ≈ 24 h (Millar, 2016; Nohales & Kay, 2016). The circadian rhythms in plants can usually be maintained under constant condition, such as constant light, mainly due to the self-sustaining nature of the endogenous circadian clock. The endogenous circadian clock system in plants usually consists of three parts: a complex core oscillator composed of interlocked transcriptional feedback loops, an environmental time clue input pathway, and the output pathway which delivers biorhythmic information generated by the core oscillator to the downstream activities, such as metabolism and growth (Montaigu, Tóth & Coupland, 2010). In plants, the biological events controlled by the biological clock systems involve basic physiological and biochemical processes, such as photosynthesis, leaf movement, cell growth, gene expression and stress response, etc. A complete plant circadian clock model has been established in A. thaliana. More than 20 circadian clock-associated genes have been identified in the A. thaliana genome, including MYELOBLASTOSIS (MYB)-related genes CIRCADIAN CLOCK ASSOCIATED 1 (CCA1) and LATE ELONGATED HYPOCOTYL (LHY), the PSEUDO RESPONSE REGULATORS (PRR s; including PRR5, PRR7 and PRR9), TIMING OF CAB EXPRESSION 1 (TOC1/PRR1), and the members of the evening complex; LUX ARRHYTHMO (LUX)/PHYTOCLOCK 1 (PCL1), EARLY FLOWERING (ELF3) and ELF4, etc (Calixto, Waugh & Brown, 2015). This model reveals that the A. thaliana core oscillator consists of several interlocked transcriptional feedback loops. Under normal conditions, the protein expression products of the early morning genes CCA1 and LHY inhibited the expression of the night gene TOC1 (PRR1) at dawn, whereas the accumulation of TOC1 expression products inhibited the expression of CCA1 and LHY genes in the night. These genes and their expression products form a negative feedback control loop and serve as the core oscillator of the A. thaliana circadian clock system. CCA1/LHY can also form a morning negative feedback loop with PRR9/PRR7, two members of the PRR family. Acting as a transcriptional factor, the heterodimer of CCA1 and LHY activates the transcription of PRR9 and PRR7 (Capovilla et al., 2015; Park, Seo & Park, 2012). In turn, the accumulated PRR9/PRR7 proteins directly inhibit the transcription of the CCA1 and LHY genes, and also alleviate the inhibition of the TOC1 transcription caused by CCA1/LHY. Accordingly, this negative feedback loop drives the plant circadian clock system, by shifting from the morning phase to the afternoon phase. Subsequently, the accumulated TOC1 protein can trigger the formation of the evening negative feedback loop by combining with the promoter regions of LUX and ELF4 that encodes two core proteins in the evening complex (Calixto, Waugh & Brown, 2015; Huang et al., 2012).

Although the molecular components of the circadian clocks in various living organisms are not conserved across species, most of the circadian clock oscillators work through a similar mechanism of transcription/translation feedback loops (Loudon, 2012). The circadian clock genes have been identified in some important cereal crops, such as rice and maize. These clock genes in monocotyledonous plants have been found to share high homology with the circadian clock genes in A. thaliana and even display similar rhythmic expression patterns. Recent genomics and transcriptomics studies in barley have confirmed the existence of orthologous circadian clock genes of A. thaliana and Oryza sativa with similar rhythmic expression patterns and amino acid sequences in barley (Calixto, Waugh & Brown, 2015; Campoli et al., 2012; Campoli et al., 2013; Deng et al., 2015; Dunford et al., 2005). These findings suggest that most components of the endogenous circadian clock system in barley possibly share same features with those reported in other cereal crops circadian clocks.

Hulless barley (Hordeum vulgare L. var. nudum hook. f.), also known as highland barley, is a cultivar of the oriental cultivated barley, which originates from and is still cultivated mainly in the regions around the Tibet-Qinghai plateau. Indeed, it has been cultivated as the most important staple crop on the Tibet-Qinghai Plateau for 3,500 years (Zeng et al., 2018). Compared with the fertile land in the low altitude plain areas with mild and humid climates, the harsh environment of the Tibet-Qinghai Plateau is challenging for the growth of most common cereal crops. Many studies have been conducted to investigate the mechanism of multiple stress-tolerances of hulless barley under various abiotic stresses, such as salinity stress, drought and high temperature stress (Torun, 2019). In addition, hulless barley whole genome and transcriptome sequencing studies have also investigated the main mechanism of tolerance to different abiotic stresses in this plant (Wei et al., 2016; Zeng et al., 2016; Zeng et al., 2015). However, we are still far from uncovering the underlying causes of why hulless barley, rather than any other better-yielding cereal crops, is better adapted to the harsh environment in the habitats with highest average altitudes on this planet, which have led to their cultivation as an irreplaceable food crop species by the local farmers in the Tibet-Qinghai regions.

In this study, to investigate the role of exogenous melatonin in the regulation of circadian rhythmic expression of circadian clock genes under cold stresses, we evaluated expression rhythm of clock genes in hulless barley seedlings either with or without exogenous melatonin under different temperature conditions.

Materials & Methods

Materials and growth condition

The hulless barley used in the study was Kunlun 12, which was a generous gift from Dr. Tao He (Qinghai University, Xining, Qinghai province, China). The seeds were first rinsed in running tap water for 10 min, and then soaked in 75% alcohol for 30 s. Subsequently, the seeds were sterilized (Cai et al., 2018) with 0.1% HgCl2 for 10 min, and thoroughly washed 5 times with plenty of sterile water. Base on some previous studies on melatonin (Han et al., 2017) and Fig. S1, the sterilized seeds were then separately soaked in 1 µM melatonin (MT) or distilled water (mock) for 12 h. The hulless barley seeds were then transplanted into 7 cm length * 7 cm wide * 10 cm height dark seedling boxes with sterile water-moistened medical gauze and a clear plastic cover. Germination was carried out in the dark at 25 °C for the following 2 d. Then, the culture condition was switched to constant light photoperiodic illumination provided by cool white fluorescent light with a density of 100 µMol m−2 s−1 for a week. Subsequently, the seedlings were randomly divided into three groups and transplanted to individual growth chambers at various temperatures of 25 °C (Normal), 15 °C (Low temperature) or 5 °C (Cold) and the same 100 µMol m−2 s−112 h/12 h light/dark photoperiodic illumination condition for 2 d. To avoid the effects of diurnal light changes, the illumination for all the growth chamber was then switched to 100 µMol m−2 s−1constant light by cool white fluorescent light during the following experimental procedure. According to different processing conditions, the experimental materials were divided into the following 6 groups: 25 °C group, 25 °C+MT group, 15 °C group, 15 °C+MT group, 5 °C group and 5 °C+MT group. A more detail description is given in Fig. S2. Starting from the time point at which the light was on (Circadian Time, CT0), about 100 mg leave tissues were harvested every 4 h continuously during the following 3 days. All the plant samples, isolated at each time point for subsequent analysis, were immediately stored at −80 °C after a brief deep freezing in liquid nitrogen.

Effect of exogenous melatonin on the germination rate of hulless barley seeds

After treatment as described above by 1 µM melatonin or distilled water, the hulless barley seeds were randomly divided into mock or MT group. The seed germination rates of the different tests were determined at 25 °C.

Determination of the effects of melatonin treatment on plant vegetative growth

After treatment as described above by 1 µM melatonin or distilled water, the seedlings were randomly divided into three groups and transplanted to individual growth chambers at various temperatures of 25, 15 or 5 °C and the same 100 µMol m−2 s−112 h/12 h light/dark photoperiodic illumination condition for 5 d. Then, their morphological indices, such as root length, leaf length, fresh or dry weight of each plantlet were measured.

Determination of endogenous melatonin contents in hulless barley seedlings

An acetone–methanol method of Pape & Lüning (2006)) was used to assay endogenous melatonin contents in 1.0 g leaf tissue from each individual seedling that harvested at CT0 and CT72. First, the leaf tissue sample of the hulless barley was fully homogenized to a fine powder in liquid nitrogen with a mortar and pestle. Then, 5 ml of extraction buffer (mixture of acetone, methanol and water at a volume ratio of 89:10:1) was added to the powder, and the grinding of the tissue homogenate was continued on ice for 5 min in the dark. Subsequently, the supernatant was collected by centrifugation at 4,500 g for 5 min in a refrigerated centrifuge and transferred to a new centrifuge tube containing 0.5 mL of 1% trichloroacetic acid (TCA) to precipitate the protein. Ultimately, the melatonin content in the supernatant from the different groups of hulless barley seedlings was determined using an Extraction-Melatonin ELISA kit (RE 54021; IBL International GmbH, Hamburg, Germany). To avoid the degradation of melatonin by natural light, all the experiments involving this compound were conducted in a dark room.

Quantitative real-time PCR (qPCR) analysis and rhythm analysis of circadian clock genes in hulless barley seedlings

Total RNA from plant tissue samples was extracted with an RNAiso Plus kit (Takara, Dalian, China). Then, the extracted RNA samples were used to as templates for the subsequent reverse transcription reactions carried out using a PrimeScript™ RT reagent Kit with gDNA Eraser (Takara). The synthesized cDNAs were purified using the MiniBEST DNA Fragment Purification Kit Ver. 4.0 (Takara), and stored at −80 °C freezers until the subsequent qPCR analyses were performed.

Using primers specific for the different circadian clock genes listed in Table S1 in supplement file, all the qPCR reactions were performed on a CFX96 Real-Time PCR Detection System (Bio-Rad Laboratories Inc., Hercules, CA, USA). A TB Green Premix Ex Taq™ II kit (Takara) was used to perform the amplifications by a two-step method. All samples were first pre-denatured at 94 °C for 30 s, followed by 40 cycles of denaturation at 94 °C for 5 s, annealing and elongation at 56 °C for 30 s with a read of fluorescence intensity from SYBR™ Green at the end of elongation process in each reaction. The Cq value for the qPCR amplification of individual samples is expressed as the mean value ± standard deviation (SD) from 3 independent biological replicates.

The Cq value data from 72 h time-course plant samples was uploaded to the online platform for data sharing and period analysis, BioDare2, and processed by a Spectrum Resampling (SR) method with a threshold of period length ranged from 18 to 34 h and a RAE (relative amplitude error) value (http://www.biodare.ed.ac.uk/) (Ramos-Sánchez et al., 2017; Zielinski et al., 2014). All the samples series with a feedback period data out of this setting threshold after the online processing, were screened as arrhythmia by BioDare2 automatically. Detailed circadian period and RAE (relative amplitude error) analysis results were shown in Figs. S3 and S4.

Determination of chlorophyll and carotenoid content in leaves of hulless barley under different temperature conditions

Fresh leaf tissue samples from hulless barley seedlings in an amount of 0.1 g,were harvested in a set of time-course experiments at CT0, CT12,CT 24, CT36, CT48,CT60 and CT72. Chlorophyll (Chl) and carotenoid contents in these leaf tissue samples were measured following the method of Lichtenthaler and Wellburn with slight modifications (Lichtenthaler & Wellburn, 1983). After homogenization in 20 mL of 80% acetone (v/v) using a pre-cooled mortar and pestle, the homogenate of the leaf tissue was centrifuged at 8,000 g for 10 min. Then, three mL of the supernatant from each plant tissue sample was taken to measure the absorbance at the wavelengths of 663, 646 and 470 nm on a UV/VIS T6 spectrophotometer (Persee Analytics, Beijing, China). The contents of chlorophyll and carotenoid were calculated using the Lichtenthaler and Wellburn formula (Lichtenthaler & Wellburn, 1983) and expressed as mg g−1FW.

Determination of free proline, soluble sugars and malondialdehyde (MDA) contents in leaves of hulless barley seedlings under different temperature conditions

Proper amounts of leaf tissues of hulless barley seedlings under different temperature conditions were harvested, and used to determine the intracellular levels of free proline, soluble sugars and MDA. The free proline in leaf tissues was extracted and determined using the Mahdavian’s method (Mahdavian, Ghaderian & Schat, 2016) with minor modifications. Briefly, 0.5 g of hulless barley seedling leaves was thoroughly homogenized in five mL of 3% sulfosalicylic acid solution (m/v), and centrifuged at 8,000 g for 10 min. Then, two mL of the supernatant were transferred into a test tube containing a mixture of two mL glacial acetic acid and two mL of acid ninhydrin, and the test tube was incubated in a water bath at 95 °C for 1 h. Subsequently, the red-colored reaction products were extracted and purified with four mL of toluene. The optical density (OD) value of the recovered toluene phase solution was measured at the wavelength of 520 nm. The proline content of each plant sample was determined using a standard curve for proline and expressed as µmol g−1 FW.

The soluble sugars content was measured using the anthrone method (Azarmi et al., 2015). Precisely, 0.5 g of the leaf tissues was harvested and thoroughly homogenized in five mL of 80% ethanol and centrifuged at 8,000 g for 15 min. Then, 0.1 mL of the supernatant was accurately aspirated and mixed with three mL of anthrone. The reaction mixture was incubated in a boiling water bath for 30 min and then cooled on ice. The OD625 value was recorded for each plant samples and used to calculate the soluble sugars content using a standard curve of soluble sugars reagent and expressed as mg g −1 FW.

The MDA content of each hulless barley leaf tissue sample was determined using the thiobarbituric acid assay (Nahar et al., 2015). Briefly, the leaf tissue (0.5 g) was thoroughly homogenized in 5 ml of TCA, and centrifuged at 8,000 g for 10 min. Then, 2 ml of the supernatant was aspirated and mixed with 2 ml of a 0.5% thiobarbituric acid solution. The reaction mixture was incubated in a 95 °C water bath for 30 min, and then immediately cooled on ice. After centrifugation at 5,000 g for 10 min, a supernatant aliquot was taken to measure the absorbance at 450, 532 and 600 nm. The MDA content was calculated according to the method reported by Nahar (Nahar et al., 2015) and expressed in µmol g−1 FW.

Statistical analysis

All growth data were subjected to one-way analysis of variance (ANOVA) using the SigmaStat V3.5 software (Systat Software, San Jose, CA, USA). Duncan’s multiple range test was used to assess the difference between treatments at a significance level of P < 0.05. For gene expression analysis, data analyses involving multiple comparisons were performed using Student’s t-test and the SPSS 22.0 software (SPSS Inc., Chicago, IL, USA).∗P <  0.05,∗∗P <  0.01.

Results

Seed germination and growth assay

The low temperature stress inhibited the growth of hulless barley seedlings, while melatonin pretreatment alleviated this inhibition to a certain extent (Fig. 1). At first, the germination rate of hulless barley seeds was 57% at 25 °C conditions (Fig. 2A). The germination rate analysis was carried out in our laboratory at Xi’an, Shaanxi, China with a location of 108.96° East longitude and 34.28° North latitude, an altitude of 405 m and an air pressure of 97 KPa. When the hulless barley seeds were pretreated with 1 µM exogenous melatonin, the germination rate was significantly increased to 63% (Figs. 2A–2B).

Figure 1 Growth status of hulless barley seedlings under different treatment conditions.

25 °C 25 ° C+MT, 15 °C 15 °C+MT, 5 °C and 5°C +MT.

Figure 2 Effects of exogenous melatonin on growth.

(A) Germination rate of hulless barley seeds, (B) melatonin content, (C) length of primary leaf, (D) length of root, (E) fresh weight of seedlings, (F) dry weight of seedlings. Different lowercase letters in all figures indicate significant difference at P < 0.05 level. Data are expressed as the means ±  SEM.

We also determined the effects of exogenous melatonin on leaf length, primary root length, fresh weight and dry weight of hulless barley seedlings under different temperature conditions. The results revealed that exogenous melatonin significantly promoted the growth of leaf length and primary root length and increased fresh weight and dry weight of hulless barley seedlings (Figs. 2C–2F). In particular, under the condition of cold stress, application of exogenous melatonin markedly alleviated the suppressive effects of low environmental temperature on the vegetative growth of either the hulless barley seedlings leaves, or their roots. These results indicated that a suitable concentration of melatonin could confer significant physiological protection on the barley seedlings when they were exposed to cold stress. Meanwhile, It is interesting that the dry weight of hulless barley at 5 °C is greater than that at 25 and 15 °C, which was same as previous study in hulless barley under salt stress (Ma et al., 2018) .

The rhythmic expression pattern of the circadian clock gene of hulless barley under cold stress

The expression patterns of the circadian clock genes of hulless barley seedlings were determined at 25, 15 and 5 °C. At 25 °C, the expression of the circadian clock genes, including HvCCA1, HvTOC1, HvGI, HvLUX, HvPRR59, HvPRR73 and HvPRR95 maintained a robust circadian rhythmic profile for three days. Unexpectedly, unlike previous reports in barley and A. thaliana, the expression of HvELF3 in hulless barley exhibited an arrhythmic phenotype. At 15 °C, the expression of the morning clock gene HvCCA1 was not affected by the low temperature signals and showed a similar rhythmic expression pattern to that at 25 °C, while the total mRNA abundance of the night clock genes HvTOC1 and HvGI were significantly downregulated. In contrast, the expression levels of genes encoding key components of the evening complex (EC) of circadian clock in plants, such as HvLUX and HvELF3, were significantly upregulated. With respect to the PRR genes, although the expression levels of HvPRR59, HvPRR73 and HvPRR95 were not affected by exposure to 15 °C (low temperature), the phase of their rhythmic expression pattern was shifted forward. Consistent with expectations, the expression of all the evaluated circadian clock genes was significantly suppressed, and showed arrhythmic phenotypes under cold stress conditions at 5 °C. (Fig. 3, Fig. S3 and Fig. S4).

Figure 3 Expression pattern of circadian clock genes in hulless barley seedlings at various temperatures.

(A) HvCCA1, (B) HvTOC1, (C) HvGI, (D) HvELF3, (E) HvLUX, (F) HvPRR59, (G) HvPRR73 and (H) HvPRR95. Expression of circadian clock gene measured by RT-qPCR analysis at 25 °C (black lines), 15 °C (red lines) and or 5 °C (blue lines). Total RNA was extracted from three biological replicates. Average expression is shown relative to EF-1 α, error bars represent the standard error. Horizontal axis labels indicate circadian time (CT) after a 12h/12 h light/dark entrainment. The open rectangles under horizontal axis represent the subjective day and the gray rectangles represent the subjective night.

Effects of melatonin on the rhythmic expression pattern of circadian clock genes in hulless barley

The effects of exogenous melatonin on the rhythmic expression pattern of the circadian clock genes under various temperature conditions were evaluated after soaking the seeds of hulless barley in a 1 µM melatonin solution. The results showed that at 25 °C, exogenous melatonin remarkably induced the expression level of HvCCA1, the crucial morning clock gene of the plant endogenous circadian clock, to nearly 30-fold higher than that of untreated mock seedlings. Additionally, exogenous melatonin also slightly inhibited the expression level of HvTOC1, which is the key gene in the evening loop of the plant circadian clock. PRR genes in hulless barley displayed diverse responses to the exogenous melatonin treatments. The expression of HvPRR73 was induced by 2∼4 folds, while that of HvPRR59 and HvPRR95 was significantly inhibited by 1 µM melatonin. Regarding other evening loop genes, HvELF3 expression was slightly induced by exogenous melatonin, but the expression of HvLUX and HvGI was suppressed by melatonin to various degrees. In general, exogenous melatonin was found to affect the rhythmic expression patterns of the circadian clock genes by influencing their amplitudes in various degrees at 25 °C. As well as, exogenous melatonin had influenced the period length of clock gene expression at 25 °C in different manners. As shown in Fig. S3, exogenous melatonin treatment significantly decreased the period length of HvTOC1 (about 2.5h), while remarkably increased the period length of HvPRR59 (2.7h) and HvGI (1.9 h). (Fig. 4, Fig. S3 and Fig. S4).

Figure 4 Expression of circadian clock genes in hulless barley seedlings treated with 1 μM melatonin at 25 ° C.

HvCCA1, (B) HvTOC1, (C) HvGI, (D) HvELF3, (E) HvLUX, (F) HvPRR59, (G) HvPRR73 and (H) HvPRR95. Plots of measured expression profiles of circadian clock genes for the 25 °C group (black lines) and 25 °C+MT group (red lines) hulless barley seedlings. Total RNA for the analysis of the clock genes was extracted from three biological replicates. Average Cq values from three independent RT-qPCR amplifications were used to determine the relative expression levels of target genes after calibration with a reference gene, EF-1 α. Error bars represent standard errors. Asterisks indicate significant differences in different treatment groups (∗P < 0.05, ∗∗P < 0.01).

Under 15 °C (low temperature) conditions, the expression levels of the HvCCA1, HvPRR59, HvPRR95, HvTOC1, HvLUX and HvELF3 genes in hulless barley seedlings that had been pretreated with 1 µM exogenous melatonin, were substantially suppressed compared with those of the seedlings at 25 °C. Among all tested circadian clock genes, the rhythmic expression of some genes was not affected, such as those of HvPRR59 and HvPRR95. However, the peak phase of some others was delayed, such as HvPRR73 and HvGI, whereas the peak phase of the evening loop genes HvTOC1 and HvLUX was shifted forward. The circadian rhythmic expression of morning gene HvCCA1 was completely disrupted and showed an arrhythmic pattern in melatonin pretreated hulless barley seedlings (RAE >0.6). These results indicated that exogenous melatonin inhibited the expression of most circadian clock genes and influenced the phase of some other clock genes under low temperature conditions. Additionally, these results also suggested that the night clock genes (such as HvTOC1 and HvLUX) were the potential targets for melatonin pretreatment at 15 °C. Interestingly, the period of all rhythmically expressed gene was extended by exogenous melatonin at 15 °C (Fig. 5, Figs. S3 and S4).

Figure 5 Expression of circadian clock genes in hulless barley seedlings treated with 1 μM melatonin at 15 °C.

HvCCA1, (B) HvTOC1, (C) HvGI, (D) HvELF3, (E) HvLUX, (F) HvPRR59, (G) HvPRR73 and (H) HvPRR95. Plots of measured expression profiles of circadian clock genes for the 15 °C group (black lines) and 15 °C+MT group (red lines). Total RNA for the analysis of the clock genes was extracted from three biological replicates. Average Cq values from three independent RT-qPCR amplifications were used to determine the relative expression levels of target genes after calibration with a reference gene, EF-1 α. Error bars represent standard errors. Asterisks indicate significant differences in different treatment groups (∗P < 0.05, ∗∗P < 0.01).

Under 5 °C (cold stress) conditions, the expression of the HvPRR73, HvPRR59, HvPRR95, HvLUX, HvELF3 and HvGI genes was strongly suppressed by 1 µM melatonin treatments and showed arrhythmic phenotype in their time course expression patterns. Remarkably, exogenous melatonin treatment was able to restore the rhythmic expression of the HvCCA1 and HvTOC1 gene to some extent, although their amplitudes were much lower than the 25 °C group. It was also worth noting that the circadian oscillation of HvCCA1 and HvTOC1 peaked at the subjective night phase under exogenous melatonin treatment. These results indicated that under cold stress, the appropriate concentration of exogenous melatonin treatment could restore the rhythmicity of the expression of the core circadian clock genes that encode the pacemaker of plant circadian clock in hulless barley (Fig. 6, Figs. S3 and S4).

Figure 6 Expression of circadian clock genes in hulless barley seedlings treated with 1 μM melatonin at 5 °C.

HvCCA1, (B) HvTOC1, (C) HvGI, (D) HvELF3, (E) HvLUX, (F) HvPRR59, (G) HvPRR73 and (H) HvPRR95. Plots of measured (solid lines) expression profiles of circadian clock genes for the 5 °C group (black lines) and 5 °C+MT group (red lines). Total RNA for the analysis of the clock genes was extracted from three biological replicates. Average Cq values from three independent RT-qPCR amplifications were used to determine the relative expression levels of target genes after calibration with a reference gene, EF-1 α. Error bars represent standard errors. Asterisks indicate significant differences in different treatment groups (∗P < 0.05, ∗∗P < 0.01).

Figure 7 Concentration of chlorophyll, Carotenoids, MDA, soluble sugar and proline in hulless barley leaf tissues from the 25 °C 25 °C+MT, 15 °C 15 °C+MT, 5 °C and 5 °C+MT groups.

(A-C) chlorophyll, (D-F) Carotenoids, (G-I) MDA, (J-L) soluble sugar and (M-O) proline. Asterisks indicate significant differences in different treatment groups (∗P < 0.05, ∗∗P < 0.01). Data are expressed as the mean ± SEM (n = 3).

Determination of the chlorophyll and carotenoid contents in leaves of hulless barley

Under conditions at 25 °C, the treatment with 1 µM melatonin reduced the chlorophyll and carotenoid contents in the leaves of hulless barley seedlings at most time points, though an unexpected increase of these two photosynthetic pigments by 44% and 35% more than the mock seedlings was observed at CT48 in a time course experiment that lasted 3 d.

At 15 °C, the accumulation of chlorophyll and carotenoids showed very similar fluctuation patterns in hulless barley. In addition, at 15 °C, the exogenous melatonin significantly increased the relative amounts of cytosolic chlorophyll and carotenoids in seedling leaves after CT36. For example, triggered by the exogenous melatonin, the accumulation of chlorophyll contents in leaves peaked at the CT36 and CT60 time points by 27% and 45%, respectively, compared to the mock seedlings.

Exogenous melatonin could also increase the contents of chlorophyll and carotenoids in the leaves of hulless barley under cold stress conditions after CT24 at 5 °C. For example, compared to 5 °C group, treatment with 1 µM melatonin also significantly increased chlorophyll accumulation by 49% and 35% at CT36 and CT60, respectively in a time course experiment. These results confirmed that the proper amount of exogenous melatonin contributed to provide a better protection to these key pigments of their photosynthetic systems under low temperature and cold stress conditions (Figs. 7A–7F).

Determination of the MDA, soluble sugar and free proline content in leaves of hulless barley

MDA is a stable product of membrane lipid peroxidation, which always occurs during various abiotic stress-response processes in plants. The relative content of MDA in cells has been used either as a general intracellular indicator to ealuate the damage of plant cells under stresses, or as a common physiological indicator to estimate different resistance abilities of individual plant species under various environmental stresses (Leshem, 1987). In this study, no noticeable changes in the MDA contents of stressed hulless barley leaves under both 15 °C and 5 °C conditions were found when compared to that of the seedlings under 25 °C (Figs. 7G–7I). Nevertheless, the contents of intracellular MDA also changed significantly in all the seedlings that have been treated by 1 µM melatonin. In particular, at 5 °C, the treatment with melatonin markedly decreased the production of MDA in hulless barley leaves to between 7 and 57% at different time points (Figs. 7G–7I).

The soluble sugar content of hulless barley leaves increased significantly with when the environmental temperature dropped to 15 °C or 5 °C, as shown in Fig. 7. However, the treatment of 1 µM exogenous melatonin notably reduced the amount of soluble sugars in hulless barley seedlings under 15 ° C and 5 °C conditions. In particular, the soluble sugar content in the melatonin-treated seedlings decreased 26% to that of the mock seedlings under the same cold stress conditions at 5 °C. These results provided direct evidence for exogenous melatonin had enhanced the stress-resistance of hulless barley seedlings on the physiological levels under cold stress (Figs. 7J–7L).

Various environmental abiotic stresses, such as low temperature, high salt and excessive heavy metals always induced the accumulation of free proline in plant cells as a compatible osmoprotectant (Ashraf & Foolad, 2007). In this study, no significant changes in the free proline content in mock hulless barley seedlings were detected under various temperature conditions. When hulless barley seeds were pretreated with 1 µM exogenous melatonin, the free proline content in hulless barley seedlings was reduced to 53% at 15 °C, meanwhile exogenous melatonin did not significantly influence the proline content in hulless barley seedlings at 25 °C and 5 °C (Figs. 7M–7O).

Discussion

Cold stress is one of the most severe stresses that strongly inhibits the vegetative growth of hulless barley seedlings (Peppino Margutti et al., 2017). Many studies in A. thaliana (Bajwa et al., 2014), wheat (Turk et al., 2014) and Bermuda grass (Hu et al., 2016) have shown that treatment with the proper concentration of exogenous melatonin clearly alleviates the inhibitory effect of environmental low temperature and cold stress on plant growth. In some studies, the data also shows that higher concentrations of exogenous melatonin treatment would inhibit plant growth (Chen et al., 2009). Consistent with previous reports, we also found that treatment with 1 µM exogenous melatonin enhanced cold tolerance in hulless barley and significantly alleviated the plant growth inhibition under low temperature and cold stress conditions. However, the specific molecular mechanism by which melatonin promoted the growth of stressed plant has not been fully elucidated.

The circadian clock is an important system for controlling plant growth. In A. thaliana, more than one-third of genes are directly or indirectly controlled by circadian clock genes under natural growth conditions (Covington et al., 2008). In this study, at 25 °C, the expression of hulless barley circadian clock genes, namely HvCCA1, HvPRR73, HvPRR59, HvPRR95, HvGI, HvTOC1 and HvLUX displayed self-sustained robust rhythmic oscillations. When the seedlings of hulless barley were cultured at 15 °C, the circadian clock genes of hulless barley did maintain their rhythmic expression patterns following the temperature compensation mechanism, an intrinsic feature of all known circadian clock system.

Melatonin is a hormone that is ubiquitously present in many living organisms. Numerous studies have revealed the detail mechanism by which the melatonin plays a crucial role in the regulation of the mammalian circadian rhythms, such as the circadian sleep/wake rhythms (Nawaz et al., 2015). However, the function of melatonin in plants has not been completely elucidated (Arnao & Hernández-Ruiz, 2015). In this study, we found that the main role of melatonin in the regulation of the expression of circadian clock genes is to increase the relative expression of the morning loop genes, such as HvCCA1, and slightly inhibit that of the evening loop genes, such as HvTOC1 at 25 °C under constant light. This finding indicated that melatonin not only can promote the growth of hulless barley but also affect the expression level of circadian clock genes under normal growth conditions. In previous studies, overexpression of CCA1 was found to significantly accelerate the vegetative growth in A. thaliana (Lau et al., 2011). We hypothesized that the increasing and decreasing expression of morning clock genes plays an important role in the process of vegetative growth acceleration in hulless barley. Under 15 °C low temperature stress, treatment with exogenous melatonin significantly suppressed the expression of most circadian clock genes. In fact, even the rhythmic expression pattern of the morning loop gene CCA1 was completely lost and also showed substantially decreased relative mRNA levels in seedlings. Regarding the evening loop genes, such as HvTOC1 and HvLUX, they maintained their expression rhythms albeit with very dimmed amplitudes and disturbed phases, which were shifted forward at 15 °C. Consistent with our above mentioned hypothesis, no significant differences were found between the vegetative growths of the seedlings from the 15 °C group and 15 °C+MT group, when the expression of the morning gene HvCCA1 was severely suppressed and showed an arrhythmic pattern, while the evening clock genes displayed circadian expression pattern with disturbances in either amplitude and phase features in the melatonin pretreated hulless barley seedlings. Under 5 °C cold stress conditions, almost all circadian clock genes lost their circadian rhythms. However, the treatment with exogenous melatonin restored the circadian rhythm of HvCCA1 and HvTOC1 to some extent. This is consistent with reports that melatonin inhibits the daytime effects in the dinoflagellate Lingulodinium and that peaks of melatonin appear in nighttime in Chenopodium rubrum L. and green macroalga Ulva sp. (Kolár & Machácková, 2005; Kolář et al., 1997; Tal et al., 2011). When the growth of seedlings was evaluated again, the results showed that the growth of leaves in the melatonin pretreated hulless barley seedlings was significantly higher than that of 5 °C group seedlings, simultaneously with the partial restore of HvCCA1 and HvTOC1 expression rhythms with dimmed amplitude.

Photosynthetic pigments are important substances that responsible for the absorption, transmission, and transformation of light energy in plant photosynthesis (Ashraf & Harris, 2013). In this study, no significant changes in the chlorophyll and carotenoid contents were detected in hulless barley seedlings that were not treated with melatonin under low temperature and cold stress conditions, compared with the seedlings under normal conditions. Nevertheless, the measurements of photosynthetic pigments showed that melatonin treatment significantly increased the contents of chlorophyll and carotenoids under cold stress condition. These findings indicated that exogenous melatonin can protect the photosystem of hulless barley seedlings against the damages caused by cold stress by increasing the amount of chlorophyll and carotenoids in the cells.

As one of the most important osmoregulatory substances, soluble sugars in plant cells play a pivotal role in the physiological responses to cold stress in various plants and protect plant cells by increasing their cytosolic osmotic potentials, as well as decreasing the freezing point of their cell matrix (Markovskaya et al., 2010). In this study, the content of soluble sugars in the hulless barley seedlings increased significantly when they were exposed to a low temperature of 15 °C or 5 °C, but the treatment with melatonin greatly reduced the content of soluble sugars in hulless barley than that in the 15 °C and 5 °C groups.

Low temperature or cold stress induce various responses in plants, such as the accumulation of MDA in cells. MDA is the final product of the peroxidation of membrane lipids. Both MDA and soluble sugars have been widely used as indicators to assay the degree of damage caused by various stresses to the plant as well as been utilized to estimate the stress tolerance of various plants to abiotic stresses by researchers (Szabados & Savouré, 2010). However, no noticeable increase in the contents of MDA were detected among hulless barley seedlings that have been cultivated under low temperature or cold stress conditions in this study. These results indicated that, as a local plant adapted to the Tibet-Qinghai plateau, hulless barley had displayed a better resistance to cold stresses than other main cereal crops grown in lower altitude regions. These facts that the treatment with exogenous melatonin could reduce the contents of MDA and soluble sugar under cold stress conditions, also indicated that exogenous melatonin had alleviated the damage to hulless barley seedlings caused by cold stress.

Conclusions

In summary, under cold stress conditions, treatment with exogenous melatonin can enhance plant resistance to cold stress by regulating the circadian rhythms of the expression profiles of clock genes and accumulation profiles of some key physiological indicators in hulless barley. In addition, melatonin can attenuate the inhibition of plant growth induced by environmental cold stress. Our results indicate that melatonin is an important plant growth regulator, which can be used to improve cold stress tolerance in plants in agricultural production and thus has great application prospects.

Supplemental Information

Supplemental Information 1 Supplemental table and figures

Click here for additional data file.

Supplemental Information 2 Data

Click here for additional data file.

Additional Information and Declarations

Competing Interests

Author Contributions

Data Availability

The authors declare there are no competing interests.

Tianliang Chang and Yuwei Zhao conceived and designed the experiments, analyzed the data, prepared figures and/or tables, authored or reviewed drafts of the paper, and approved the final draft.

Yi Zhao performed the experiments, prepared figures and/or tables, and approved the final draft.

Hongyan He performed the experiments, authored or reviewed drafts of the paper, and approved the final draft.

Qianqian Xi analyzed the data, prepared figures and/or tables, and approved the final draft.

Jiayi Fu conceived and designed the experiments, authored or reviewed drafts of the paper, and approved the final draft.

The following information was supplied regarding data availability:

The raw measurements are available in the Supplemental Files.

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
