# Peer review of "Exogenous melatonin improves growth in hulless barley seedlings under cold stress by influencing the expression rhythms of circadian clock genes"

_PeerJ, doi:10.7717/peerj.10740_

## Round 0.1 · original submission · Major Revisions

Our reviewers have several suggestions to improve your paper. Please pay special attention to the important remarks by reviewer #2.

Reviewer 1 ·

Basic reporting

The authors investigated the possible mechanism by which melatonin pretreatment can promote the growth of hulless barleyunder cold stress conditions by studying the effect of melatonin on the rhythm of the circadian clock system and some physiological indicators. The work appears to have been carefully done. But there are still some revision needed.

Experimental design

(1) Please explain why you use the 1μM melatonin, why do not use melatonin at other concentration to compare the effect of melatonin on the growth of plants?
(2) Please detected the concentration of melatonin after treatment.

Validity of the findings

Topic is interesting and fit for the aims and scope for Peer J, and it’s had practical production value.

·

Basic reporting

The English is good. The structure of the papers is correct, but the data interpretation is misleading.

Experimental design

The methods are appropriate.

Validity of the findings

The data are sound, but the conclusions do not reflect some of the data.

Additional comments

The paper discusses melatonin mechanisms promoting growth of hulless barley under cold stress were investigated. The application of 1 μM melatonin alleviates the growth inhibition of hulless barley seedlings caused by cold stress. Exogenous melatonin also restores circadian rhythmic oscillation of circadian clock genes, such as HvCCA1 and HvTOC1 and reduces accumulation of key physiological indicators under cold stress, including malondialdehyde (MDA) and soluble sugars.

The introduction is detailed and informative and provides good background to present study. However, the results are complex and the interpretation of the circadian results needs more careful consideration. At all temperatures, melatonin has different effects on the various circadian clock associated genes. Both the control and the melatonin data at 5 deg C (Fig. 6) are quite variable and do not show clear periodic trends. Even at 15 deg C the melatonin data seem less periodic than the control. So, the title of the paper does not correspond to the results. The data trends of chlorophyll, carotenoids, MDA, soluble sugars and proline contents at different temperatures and with melatonin are interesting.

It is interesting that the dry weight of plants at 5 deg C is greater than that at 25 and 15 deg C (Fig. 2). Authors, please, comment.

Minor changes: Line 48: “temperature gradients” implies that the temperatures were changed at the time of the experiment.

Lines 196, 236, 250: “section 2.1, section 2.6” There are no numbered sections in the MS.

Fig. 2 Parts C – F would be more informative presented in the same way as part B

Reviewer 3 ·

Basic reporting

The English language could be improved to inrease the quality of this manuscript.
I feel that some parts of this paper shall be reorganized, for instance, if I understand correct, the Materials and growth condition part of the Methods and Materials seems overlap with the description of 'Effect of exogenous melatonin on the germination rate of hulless barley seeds' and other places. In the abstract, the authors repeated a statement of the general study purpose 'In this study, we investigated the possible mechanism...' in the results part, which seems better to start directly' the results showed/revealed....In the end of introduction, the authors wrote that 'The protective effects of exogenous melatonin against ... were also detected' which seems a result.
This study used 1 μM melatonin for treatment, however, higher concentration may inhibit the plant growth while a low concentration of melatonin may promote the growth, as found our in a related study (Chen et al., Journal of Plant Physiology 166: 324-328) . the authors may want to discuss the implication of that finding for their results.

Experimental design

The study used three temperature treatments of 25 ℃(Normal), 15 ℃(Low temperature) or 5 ℃(Cold) and each was incubated in each of three growth chambers, however, sometimes the authors mentioned room temperature of 25 ℃.
I understand that there is no repeat of treatment for temperature due to the actual experimental condition/facilities; the authors may want to mention their limited findings due to limited experimental design.

Validity of the findings

This study had provided evidences for the melatonin in alleviating the cool stress response in hulless barley and proved the response by gene expression evidences.
The authors may want to provide reason(s) why they apply only a concentration of 1 μM for Melatonin treatment, if it is by chance or by other purpose?

Additional comments

Not sure why legends of each figure always repeat the first sentence, I guess it could be the format request by the journal? The legend for Figure 7 seems unfinished.

·

Basic reporting

Line 56 is missing a space between "barley" and "under".
Line 182, "and" after "water" should be changed to "or".
Line 187. It should be the effect of melatonin treatment on plant vegetative growth.
Line 196. Which title is section2.1, the expression here is not very clear
Line 236. "section 2.1" the same problem as above
Line 250. "section 2.6"the same problem as above
Line482-493. There should be a picture of the physiological indicators of untreated barley at different temperatures, which can reflect the original excellent resistance of hulless barley and the powerful regulation ability of melatonin.
The suppose in the discussion is wonderful and can explain the experimental conclusions to a certain extent, but there is a lack of experimental evidence. If it can be further proved, it must be an excellent article.

Experimental design

In the content of lines 165-167, what is the meaning of continuous light for a week?
Wang et al. (Plant Cell & Environment. 2020; 43: 637– 648 ) studied the influence of light and temperature on soybean circadian rhythm, they used a 14d 12/12 day-night ratio to Entrainment the soybean materials, and there was no 7 days‘ continuous light before that.

Line 317-318. The seeds were soaked with MT 12 hours, germinate 2 days, continuous light for 7 days, Entrainment for 2 days, and then 3 days of continuous light for sampling to determine circadian gene expression. To what extent can these explain the results are the effect of melatonin on the biological clock genes?
Suggestions: 1. Construct different melatonin overexpression and interference hulless barley strains, and obtain melatonin differential strains for relevant experiments to illustrate the effect of melatonin on the circadian rhythm.
2. After Entrainment, spray melatonin to determine the biological clock-related genes.

Validity of the findings

no comment

Additional comments

Rhythm genes are mainly regulated by light, then by temperature. After these, it is melatonin that comes into play.
This article combines temperature and melatonin, which is a good starting point. But it feels that the combination is not very well, the experimental design is not ideal. In addition, the fluorescence quantification is only related data, and there is no direct causal evidence.

---

## Round 0.2 · Minor Revisions

Please address the few remaining issues.

Reviewer 1 ·

Basic reporting

The author addressed all questions, and I agree to accept the MS.

Experimental design

No question.

Validity of the findings

No question.

Additional comments

No question.

·

Basic reporting

The English needs some attention:

For instance:

Line 48: “there is…” change to “there are…”
Line 153: “to instigate” change to “to investigate”
Line 418: “In some studies, It also shows…” change to “ In some studies, the data also show…”

Experimental design

The experimental design is satisfactory.

Validity of the findings

The findings are interesting and novel.

Additional comments

The authors improved the MS along my suggestions.

I still think that the title is misleading, as exogenous melatonin seems to have different effects on different clock genes at different temperatures. I suggest more general title, such as:

Exogenous melatonin improves growth in hulless barley seedlings under cold stress by influencing the expression rhythms of circadian clock genes.

---

## Round 0.3 · Minor Revisions

I have a final request for you: there is a need for better identification of the genes associated with circadian rhythm and the PCR primers associated in the assay study. A listing of the reference genes used and the primers used need to be included. Other than this the manuscript was well organized and read cleanly.

---

## Round 0.4 · accepted · Accept

Thank you for addressing the last issues!